# Genome-Wide Identification and Expression Analysis of Cytokinin Response Regulator (RR) Genes in the Woody Plant *Jatropha curcas* and Functional Analysis of *JcRR12* in *Arabidopsis*

**DOI:** 10.3390/ijms231911388

**Published:** 2022-09-27

**Authors:** Xianchen Geng, Chun Zhang, Lida Wei, Kai Lin, Zeng-Fu Xu

**Affiliations:** 1State Key Laboratory for Conservation and Utilization of Subtropical Agro-Bioresources, College of Forestry, Guangxi University, Nanning 530004, China; 2Key Laboratory of National Forestry and Grassland Administration for Fast-Growing Tree Breeding and Cultivation in Central and Southern China, College of Forestry, Guangxi University, Nanning 530004, China

**Keywords:** bioinformatics, cytokinin, *Jatropha curcas*, plant development, response regulator

## Abstract

The cytokinin (CK) response regulator (RR) gene family plays a pivotal role in regulating the developmental and environmental responses of plants. Axillary bud outgrowth in the perennial woody plant *Jatropha curcas* is regulated by the crosstalk between CK and gibberellins (GA). In this study, we first analyzed the effects of gibberellin A3 (GA_3_), lovastatin (a CK synthesis inhibitor), decapitation, and their interaction, on the outgrowth of axillary buds. The results indicate that lovastatin completely inhibited GA-promoted axillary bud outgrowth and partially weakened the decapitation-promoted axillary bud outgrowth. To further characterize and understand the role of CK signaling in promoting the development of female flowers and branches, we performed bioinformatics and expression analyses to characterize the CK RR gene (*JcRR*) family in *J. curcas*. A total of 14 members of the JcRR family were identified; these genes were distributed on 10 chromosomes. Phylogenetic analysis indicated that the corresponding RR proteins are evolutionarily conserved across different plant species, and the Myb-like DNA-binding domain divides the 14 members of the JcRR family into type-A and type-B proteins. Further analysis of *cis*-acting elements in the promoter regions of *JcRRs* suggests that *JcRRs* are expressed in response to phytohormones, light, and abiotic stress factors; thus, *JcRRs* may be involved in some plant development processes. Genomic sequence comparison revealed that segmental duplication may have played crucial roles in the expansion of the JcRR gene family, and five pairs of duplicated genes were all subjected to purifying selection. By analyzing RNA sequencing (RNA-seq) and quantitative reverse transcription-polymerase chain reaction (qRT–PCR) data, we characterized that the temporospatial expression patterns of *JcRRs* during the development of various tissues and the response of these genes to phytohormones and abiotic stress. The *JcRRs* were mainly expressed in the roots, while they also exhibited differential expression patterns in other tissues. The expression levels of all six type-A and one type-B *JcRRs* increased in response to 6-benzylaminopurine (6-BA), while the four type-B *JcRRs* levels decreased. The expression levels of two type-B *JcRRs* increased in response to exogenous GA_3_ treatment, while those of three type-A and three type-B *JcRRs* decreased. We found that type-A *JcRRs* may play a positive role in the continuous growth of axillary buds, while the role of type-B *JcRRs* might be the opposite. In response to abiotic stress, the expression levels of two type-A and three type-B *JcRRs* strongly increased. The overexpression of *JcRR12* in *Arabidopsis thaliana* slightly increased the numbers of rosette branches after decapitation, but not under normal conditions. In conclusion, our results provide detailed knowledge of *JcRRs* for further analysis of CK signaling and *JcRR* functions in *J. curcas*.

## 1. Introduction

Cytokinins (CKs) are kinds of phytohormones that participate in various growth and response processes during plant development. Many studies have found that CKs are beneficial for cell division [1,2,3], development of plant organs [4,5,6,7] and responses to environment stimuli [8,9,10]. CK signals are transmitted through a complex two-component system (TCS), in which histidine kinases (HKs) are auto-phosphorylated after combining with CKs and then transfer phosphate groups from histidine in the kinase region to aspartate in the signal-receiving region. Next, the phosphate groups on aspartate are delivered to histidine-phosphotransferase proteins (HPs) in the cytoplasm, and the phosphorylated HPs enter the nucleus and transfer the phosphate groups to downstream response regulators (RRs) [11]. Functional RRs mainly include type-A and type-B RRs; the former is involved in negative regulation to balance CK signals, while the latter has a positive effect on activating downstream CK-responsive genes [12]. Type-B RRs contain a conserved receiver domain at the N-terminus and a variable-length fragment at the C-terminus, which contains a conversed Myb-like DNA-binding domain that functions in DNA binding and the activation of downstream gene transcription [13]. The (A/G) GAT core sequence corresponds to a critical site for the binding of type-B RRs, which is sufficient for the interaction with other phytohormones and transcription factors [14]. Type-A RRs lack this DNA-binding domain, so they do not affect the transcription of CK response genes directly. However, they negatively regulate CK signaling via competition with type-B RRs for phosphorylation signaling to weaken CK transcriptional activity [15], which is vital for optimal plant growth and development.

RR family members have been identified and functionally characterized in different plant species [16,17,18,19,20]. Studies have found that there are crucial roles of RRs in plant development and growth. In Arabidopsis, type-B ARR12 promotes regeneration, while ARR1 inhibits regeneration in an ARR12-dependent manner [21]. ARR1 inhibits axillary bud outgrowth in Arabidopsis [22], whereas SlRR10, a homolog of ARR1, promotes axillary bud outgrowth in tomato [23]. This means that the functions of different homologous RRs vary in diverse species. In rice, a monocot, the type-B *OsRR22-*overexpressing lines have smaller panicles and reduced branching [24]. Overexpression of the type-B *OsORR2* gene has been shown to decrease plant height [25]. In Populus, the type-B *PtRR13* gene negatively regulates adventitious root development [26]. The type-A *ARR7* and *ARR15* genes alter embryonic root patterns via interactions with auxin in Arabidopsis [27]. The function of type-B members is largely redundant [28], and *arr1 arr10 arr12* triple mutants display most hallmarks of CK signaling deficiency symptoms and enhanced tolerance to drought stress [29]. A recent study found that the overexpression of the type-A *ZmRR1* gene confers chilling tolerance to maize [30]. Taken together, the results of the above studies imply that *RR* genes also play a role in abiotic stress resistance, and studies of their regulatory networks are also important for plant breeding.

*Jatropha curcas* is a perennial woody plant species whose seeds contain oil, and this species was once considered a promising raw material for biofuels [31,32,33]. Low yields are a major concern for the popularization and application of this species [34,35,36,37], whereas reasonably increasing the number of female flowers and branches could solve this problem. In previous studies, we found that CK plays important roles in flower differentiation and axillary bud outgrowth [38,39,40]. This study will help us to further illustrate how CK signaling affects these processes. Moreover, interactions among plant hormone signaling pathways constitute effective means for hormones to regulate the activity of each other, and the study of JcRRs is highly important for in-depth exploration of the interactions between CK and other plant hormones. A recent de novo genome assembly and Hi-C analysis have significantly improved our abilities to understand the structures and functions of the *JcRR* gene family members involved in CK signal transduction [41]. To identify potential RR members and better understand the evolution and function of *RR* genes in *J. curcas*, genome-wide identification and analysis of *JcRRs* were conducted in this study. We identified 14 *RR* genes in *J. curcas*, namely, 6 type-A and 8 type-B *JcRRs*. Using transcriptome data and quantitative reverse transcription-polymerase chain reaction (qRT–PCR) analysis, we determined that the expression profiles of these genes in different tissues and organs, and evaluated the expression of these genes in response to exogenous 6-benzylaminopurine (6-BA) treatment [42].

## 2. Results

### 2.1. CK Is Required for GA- and Decapitation-Promoted Axillary Bud Outgrowth in J. curcas

To further analyze the crosstalk between CK and other factors (gibberellins (GA), auxin, and sugar) in *J. curcas* seedlings during the outgrowth of axillary buds, we treated the seedlings with gibberellin A3 (GA_3_), decapitation and lovastatin (Lov) (an inhibitor of cytokinin biosynthesis). The length of the axillary buds were measured and calculated at 14 d after treatment. As shown in Figure 1, GA_3_ and decapitation treatment promoted the outgrowth of axillary buds, while the latter had a stronger effect. We then investigated whether the inhibition of endogenous CK biosynthesis could affect this process. Accordingly, we found that the elongation of the stimulated axillary bud was significantly reduced by Lov treatment. In addition, the GA_3_ + Lov treatment had no effect on the outgrowth of axillary buds, showing no significant difference compared with the mock treatment, which indicates that CK played a critical effect on GA-promoted axillary bud outgrowth. In addition, Lov treatment partially reduced the length of axillary bud outgrowth, which indicated that CK also had an important impact on the release of apical dominance.

### 2.2. Identification of the RR Gene Family in J. curcas

Because all the *RR* members in Arabidopsis, Populus and rice had been identified, and their functions had been characterized, their sequences were selected to query the *J. curcas* genome, and we first obtained 40 candidates after querying. To identify the RR proteins encoded in the genome of *J. curcas*, local BLAST searches via the hidden Markov models (HMMs) of the SMART and Pfam databases were conducted. Then we performed a structural domain screening based on the Pfam database. All of the predicted *JcRRs* shared the same conserved D-D-K receiver domain; furthermore, the type-B JcRRs contained an additional Myb-like DNA-binding domain. Finally, 14 JcRRs members were retained according to the conserved domain of RR within the *J. curcas* genome (Table 1). There were significantly fewer JcRRs than Arabidopsis RR homologs (21 ARR members), which indicated that the regulatory network and function of RRs in *J. curcas* may be more flexible and complex. To better compare the functional differences between *JcRRs* and ARRs, we named these *JcRRs* according to their homologs in Arabidopsis.

We constructed a phylogenetic tree on the basis of the results of an alignment of RR protein sequences from *J. curcas*, the dicotyledonous model plant *Arabidopsis thaliana*, the woody plant species *Populus trichocarpa*, and the monocotyledonous plant *Oryza sativa* (Figure 2 and Appendix A). The RR proteins were clustered into four monophyletic groups. The results showed that the RR proteins of *J. curcas* are more closely related to those of the dicotyledonous species Populus and Arabidopsis than to those of the monocotyledonous species rice.

We then predicted the basic characteristics of the JcRR members, including the encoding gene ID, encoding gene type, protein length, molecular weight (MW), theoretical isoelectric point (pI) and subcellular localization (Table 1). The lengths of the JcRR proteins ranged from 158 (JcRR16) to 741 (JcRR14) amino acids (aa), the MW ranged from 17.77 (JcRR16) to 80.28 (JcRR14) kDa, and the theoretical pI values ranged from 4.86 (JcRR8) to 8.28 (JcRR7). Subcellular localization prediction analysis showed that all JcRR proteins were located in the nucleus; therefore, they can interact with various transcription factors. The grand average of hydropathicity (GRAVY) value ranged from −0.817 (JcRR8) to −0.248 (JcRR7), indicating that all the JcRR proteins were hydrophilic.

### 2.3. Gene Structure Analysis and Chromosome Locations

To further explore the structure and sequence characteristics of *JcRRs* in *J. curcas*, a simpler neighbor-joining (NJ) phylogenetic tree was constructed on the basis of the results of a classification analysis in which full-length amino acid sequences were used (Appendix A). The results of the phylogenetic analysis suggested that the 14 JcRR proteins of *J. curcas* could be divided into two types: there were 6 type-A and 8 type-B members (Figure 3A).

*Cis*-acting elements within 1500 base pairs (bp) potential promoter sequences upstream of the start codon were analyzed, and the results were visualized (Figure 3B). The promoter region of the JcRRs contained multiple *cis*-acting elements, including phytohormone-responsive (abscisic acid (ABA)-responsive and GA motif), light-responsive (AE-box and MYC), plant development-related (MYB, TATA-box and CAAT-box), and stress-responsive elements (STRE, DRE and MBS). These results reflected the potential function of JcRRs during plant development as well as in response to stress. The structures of the *JcRRs* were relatively different, showing different numbers and positions of exons and introns (Figure 3C). The numbers of introns of *JcRRs* ranged from 3 (*JcRR8* and *JcRR9*) to 7 (*JcRR11b*), the numbers of exons of *JcRRs* ranged from 4 (*JcRR8* and *JcRR9*) to 8 (*JcRR11b)* and the full genomic length of *JcRRs* ranged from 1297 (*JcRR9b)* to 6229 (*JcRR14*) bp.

### 2.4. Conserved Domain and Motif Analysis

To further reveal the diversification among *JcRRs*, the conserved domains and motifs of the JcRR proteins were analyzed. All the JcRRs shared a conserved typical D-D-K receiver domain (CDD: cd17584), and the type-B JcRRs had an additional Myb-like DNA-binding domain (CDD: pfam00249), which plays a key role in regulating the expression of CK response genes (Figure 4A). The Multiple Em for Motif Elicitation (MEME) online tool was used to predict the conserved motif composition of the JcRRs; the number of motifs varied from 3 to 7. Motifs 1 and 4 were found in all JcRRs, and they may correspond to the receiver domain. Motif 2 was present only in the type-B JcRRs and may correspond to the Myb-like DNA-binding domain. In addition, other motifs were present in different JcRRs, indicating that motifs 5–10 may vary or may have been lost throughout evolution (Figure 4B and Appendix A, and Appendix A).

### 2.5. Chromosomal Distribution and Synteny Analysis

The distribution of *JcRRs* on the chromosomes was analyzed according to their physical locations in the *J. curcas* genome. All 14 genes were distributed on 10 chromosomes in *J. curcas*, and 8 of them were at the proximate or distal ends of the chromosomes. Among these genes, *JcRR11a*, *JcRR11b,* and *JcRR11L* were adjacent to each other on chromosome 3, the phenomenon of which may have been caused by a tandem duplication event (Figure 5).

Then we analyzed the gene duplication events of *JcRRs* to elucidate the evolutionary relationships of *JcRRs*. In total, five pairs of duplicated *JcRR* genes were found, including three pairs of segmental duplication (60% of all duplicated *JcRR* genes) and two pairs of tandem duplications (40% of all duplicated *JcRR* genes) (Figure 6A and Appendix A). To further understand the evolutionary constraints acting on duplicated *JcRRs*, the non-synonymous substitution rate (Ka), synonymous substitution rate (Ks), and Ka/Ks ratio of 5 pairs of duplicated *JcRRs* were calculated (Figure 6B and Appendix A). The Ka/Ks ratio of 5 pairs was less than one, which implied that they tended to be subjected to purifying selections. To establish the orthologous relationships of *JcRRs*, we performed a collinearity analysis between *J. curcas* and three other species (Arabidopsis, Populus, and rice) (Figure 6C). The results indicated that *RRs* in *J. curcas* and Populus (27 pairs) showed a closer evolutionary relationship, followed by Arabidopsis (19 pairs) and rice (7 pairs).

### 2.6. Expression Profiles of JcRRs

According to the results of the analysis of the *cis*-acting elements within the promoters (Figure 3B), the expression level of the *JcRRs* might be affected by hormones and abiotic stresses. To further understand the functions of *JcRRs*, various transcriptome data were used to generate heatmaps, such as transcriptomic data in response to phytohormones [39], tissues [43,44,45], and abiotic stress [46,47,48] (Appendix A). *JcRRs* showed diverse responses under various treatments and conditions (Figure 7). For hormone responses (Figure 7A), the expression level of *JcRR11a* strongly increased in response to both GA_3_ and 6-BA, while the expression level of *JcRR21* slightly increased. The expression levels of *JcRR4*, *JcRR9a*, and *JcRR9b* increased in response to 6-BA but decreased in response to GA_3_. The expression levels of *JcRR11b*, *JcRR11L*, and *JcRR12* significantly decreased in response to both GA_3_ and 6-BA. During the development of seeds after pollination, the expression levels of *JcRR8*, *JcRR9a*, *JcRR11L*, *JcRR16*, and *JcRR21* first increased and later decreased. *JcRR7* and *JcRR11b* showed the highest expression levels in the mature seeds. Under drought stress (Figure 7B), the expression of *JcRR8* and *JcRR11b* increased in the leaves, while *JcRR11a* and *JcRR16* were downregulated. The expression level of *JcRR11b* increased in the roots, while *JcRR9a*, *JcRR9b*, and *JcRR11L* were downregulated. Under salt stress (Figure 7B), the expression levels of *JcRR7* and *JcRR14* increased in the leaves, while *JcRR4*, *JcRR11a*, and *JcRR16* were downregulated. The expression levels of *JcRR4* and *JcRR18* increased in the roots, while *JcRR9a* and *JcRR11L* were downregulated. *JcRR4* expression was highest in leaves under cold stress, indicating that this gene may play an important role in cold tolerance (Figure 7B). We found that the expression levels of both *JcRR11a* and *JcRR16* decreased in the leaves under drought and salt stresses, and both *JcRR9a* and *JcRR11L* were downregulated in the roots. Taken together, these results implied that these genes play potential roles in the abiotic stress response, and further study of their functions is important.

Our previous studies showed that exogenous CK treatment promoted the outgrowth of axillary buds in *J. curcas* [39,49]. To better understand the regulatory mechanism, we measured the expression level of *JcRRs* via qRT–PCR analysis after 6-BA treatment (Figure 8). The expression levels of five *JcRRs* (*JcRR4*, *JcRR7*, *JcRR9a*, *JcRR9b*, and *JcRR16*) notably increased at 3 h, while the expression of *JcRR8* began to increase at 12 h after 6-BA treatment. The expression of most type-B *JcRRs* was not induced by CK, a phenomenon that was consistent with that reported in a previous study of Populus [16]. However, the expression of *JcRR11L* began to increase at 6 h after treatment (Figure 8B), and this gene may be involved in a specific regulatory pathway. The expression levels of *JcRR11b*, *JcRR14*, *JcRR18*, and *JcRR21* slightly decreased after treatment. Taken together, these results indicated that type-A JcRRs may play a positive role in the continuous growth of axillary buds, while the roles of type-B JcRRs might be the opposite.

To determine the tissue preferences of the *JcRR* gene family members, qRT–PCR analysis was conducted to further confirm the expression levels of *JcRRs* in different tissues (Figure 9). *JcRR4* was mainly expressed in the roots, whereas *JcRR7* was mainly expressed in the roots, female flowers, and male flowers. *JcRR8* was mainly expressed in the roots, male flowers, and fruits, whereas *JcRR9a* was expressed in all tissues except the female flowers and male flowers. *JcRR9b* was mainly expressed in the roots, shoot apical meristems, axillary buds, stems, and fruits. *JcRR16* showed a high expression level only in the roots. While *JcRR1* was expressed the highest in the roots, its expression was relatively consistent in the other tissues. *JcRR11a* was mainly expressed in the roots, fruits, and seeds, whereas *JcRR11b* was mainly expressed in the roots, mature leaves, female flowers, and fruits. *JcRR11L* exhibited high expression in the young leaves and mature leaves and extremely high expression levels in the roots. *JcRR12* and *JcRR14* were expressed in all the tested tissues. *JcRR18* showed the highest expression in the seeds and relatively high levels in the roots and female flowers. Compared to the other *JcRRs*, *JcRR21* showed lower expression and was mainly expressed in the roots, stems, fruits, and seeds. In general, the expression patterns of these 14 *JcRRs* were consistent with the RNA sequencing (RNA-seq) data. In the future, researchers should consider how *JcRRs* transmit CK signals and activate downstream genes.

### 2.7. Overexpression of J. curcas JcRR12 in Arabidopsis Led to Slightly Increased Numbers of Rosette Branches after Decapitation Treatment

To investigate the functions of *JcRR12* during axillary bud outgrowth, *JcRR12* was overexpressed in *Arabidopsis* under the control of a CaMV (Capsicum Mottle Virus) 35S promoter. We obtained ten independent *35S:JcRR12* transgenic *Arabidopsis* lines (Figure 10A). After examining the expression level of each line by qRT-PCR, three lines (OE-3, OE-4 and OE-9) that expressed the highest were retained for subsequent studies (Figure 10B). During plant growth and development, we did not find any changes in the number of rosette branches between wild-type (WT) and transgenic plants under normal growth conditions. To further study the function of *JcRR12* in axillary bud outgrowth, we generated a more-branches phenotype by decapitation treatment after the onset of flowering (Figure 10C,D). We found that transgenic plants produced more rosette branches than WT plants after decapitation treatment. These results indicated that *JcRR12* was able to affect decapitation-induced axillary bud outgrowth in Arabidopsis.

## 3. Discussion

Decreased CK completely inhibited GA_3_-stimulated axillary bud outgrowth and weakened the release of apical dominance. This result indicates that GA-promoted axillary bud outgrowth is dependent on CK and that GA may interact with CK signaling to indirectly influence the expression of downstream related genes. In addition, other factors may regulate axillary bud outgrowth by interacting with CK. Therefore, we performed the identification and analysis to study the function of RR genes in *J. curcas*, which are key components of cytokinin signaling.

In this study, we identified 14 *RR* genes in *J. curcas*, and there are 21, 22 and 22 *RR* genes in Arabidopsis [50], Populus [16] and rice [51], respectively. We named JcRR members according to their corresponding homologs in Arabidopsis, which facilitated the study of their functions. There were fewer RRs in *J. curcas* than in other species, which may be due to the loss of some homologous genes during chromosome replication.

*Cis*-acting elements play an important role in gene expressions regulatory networks, such as those involved in plant growth and development [52], stress responses [53], hormone responses [54,55], and signal transduction [56]. We analyzed the promoter regions of *JcRRs* and found that many important *cis*-acting elements are present in the promoters (Figure 3B), such as various plant hormone (abscisic acid (ABA)-responsive and GA motif)-related, light (AE-box and MYC)-related and stress response (STRE, DRE, and MBS) elements. Although previous studies have partially illustrated their role in stress resistance, further studies on the interaction between CK signaling and stress stimuli are needed. Studying the crosstalk between JcRRs and different phytohormones is also valuable. In Arabidopsis, ARR1 recruits DELLA proteins, which are transcriptional coactivators that regulate the expression of CK-responsive genes [57]. We found that CK and GA synergistically regulated the axillary bud outgrowth of *J. curcas*, and CK could regulate the expression levels of GA synthetase (*JcGA20oxs*, *JcGA3oxs*) and degradase (*JcGA2oxs*) enzyme genes to maintain GA homeostasis [39]. Functional analysis of JcRRs may help to elucidate the regulatory network underlying the interaction between CK and GA signaling during axillary bud outgrowth. An analysis of conserved motifs was performed on the RR gene family proteins, and 10 motifs were identified (Figure 4B). Two conserved motifs corresponding to the D-D-K receiver domain were present in all the proteins, so they were all RR gene family members. Eight type-B members shared the same motif—motif 2 (which corresponds to a Myb-like DNA-binding domain), which was vital for activating downstream genes, while the 6 type-A members did not have this motif. In addition, other motifs were present in different JcRRs, indicating that they may vary or have been lost during evolution (Figure 4B). The numbers and distributions in gene families were determined by duplication events, which helped to explore the differences in functions among gene family members [58]. Among the 14 *JcRRs*, three gene-pairs from segmental duplication (*JcRR9b*/*JcRR9a*, *JcRR7*/*JcRR4*, and *JcRR12*/*JcRR18*) and two from tandem duplication (*JcRR11a*/*JcRR11L*, and *JcRR11L*/*JcRR11b*) were found in *J. curcas*. We found that all the duplicated gene pairs had Ka/Ks ratios of less than one, suggesting shorter divergence of time and more limited functional divergence. There are more collinear blocks of JcRR and PtRR genes, indicating a conserved evolutionary history before the divergence between *J. curcas* and Populus.

JcRRs showed diverse responses to various treatments and developmental conditions. We generated heat maps to study the response patterns of members of the JcRR gene family (Figure 7). The expression level of *JcRR12* decreased in response to GA_3_ treatment (Figure 7A), the findings of which are consistent with those of a previous study in Arabidopsis showing that GA repressed the expression of *ARR1*, which was mediated by the DELLA protein RGA [59]. The expression of *JcRR11a* was strongly upregulated in response to GA and CK (Figure 7A), suggesting that *JcRR11a* may be involved in processes regulated synergistically by GA and CK, such as axillary bud outgrowth of *J. curcas* [39,49], although CK and GA are often antagonistic regulators of multiple developmental processes [60,61]. During the development of seeds after pollination, the expression levels of *JcRR8*, *JcRR9a*, *JcRR11L*, *JcRR16,* and *JcRR21* increased at early stages and then decreased, while *JcRR7* and *JcRR11b* showed the highest expression in the mature seeds (at 45 days after pollination (DAP)) (Figure 7A); together, these findings indicated that these *JcRRs* could play roles in seed development.

To verify the reliability of the transcriptome data shown in Figure 7A, using qRT–PCR analysis, we analyzed the expression patterns of *JcRRs* in response to 6-BA. Overall, the results of qRT–PCR analysis for the type-A *JcRRs* were generally consistent with those of the transcriptome data, although this was less so for the type-B *JcRRs*. All 6 type-A and 1 type-B (*JcRR11L*) *JcRRs* were upregulated in response to 6-BA treatment, while 4 type-B *JcRRs* were downregulated (Figure 8). The expression of most type-B *JcRRs* was not induced by exogenous 6-BA treatment, which was consistent with the findings of a previous study in Populus [16]. Moreover, our previous study found that overexpressing *AtIPT4* under the control of the flower-specific *JcTM6* promoter increased the number of inflorescences and led to a high expression level of *JcRR3* (named *JcRR4* in this article) in the inflorescences [38]. In rice, *OsRR5* (a homolog of *JcRR9b*) specifically participates in strigolactone (SL)-controlled CK responses in shoot bases [62]. Since type-A *JcRRs* were continuously expressed upon 6-BA treatment, they may promote axillary bud growth and inflorescence formation, while the functions of type-B *JcRRs* might be opposite.

Under abiotic stress, 2 type-A and 2 type-B *JcRRs* were consistently downregulated (Figure 7B). Their expression changes may be related to stress resistance. *JcRR4* was hardly expressed in the leaves, and the highest expression level occurring in the leaves after cold treatment indicated that this may be vital for cold tolerance. Arabidopsis plants overexpressing *ARR5* were drought resistant [63], and accordingly, the expression level of *JcRR7* increased after drought stress. *OsRR6* (a homolog of *JcRR9a*) plays an important role in the abiotic stress response and CK signaling [64], while *JcRR9a* exhibits the same response patterns. Our results imply that these JcRRs may be important to the abiotic stress response. However, further studies are needed to elucidate their functions.

In addition, we performed a qRT–PCR analysis to further explore the expression profiles of *JcRRs* in different tissues (Figure 9). In general, CKs are mainly produced in the roots and later transported to the shoots [65], but many CKs are also produced in other tissues and organs [66]. In Arabidopsis, the spatial expression pattern of *ARR1*, *ARR2*, *ARR10,* and *ARR12* revealed a comprehensive distribution in different tissues, while those of the remaining members varied [67], and the functions of these genes in CK signaling in Arabidopsis were essential but redundant [28]. According to the comprehensive expression profiles, *JcRR1* and *JcRR12* may be involved in regulatory networks and have functions similar to those of *ARR1*, *ARR2*, *ARR10,* and *ARR12* in Arabidopsis. Although most *JcRRs* showed the highest expression in the roots, spatial expression pattern analysis showed that *JcRRs* may play various functions among different tissues, excluding their key effects on root development. *JcRR7* may regulate the development of flowers, and *JcRR8* may be involved in the development of male flowers and fruits. *JcRR9a* and *JcRR9b* may participate in cell division, therefore regulating the growth of shoot apical meristems as well as axillary buds. *JcRR11a* and *JcRR11b* may have a similar function in terms of regulating fruit development. *JcRR11L* may regulate the development of leaves, and *JcRR18* and *JcRR21* may mainly regulate the development of seeds. Taken together, the above results indicated that different *JcRRs* might be involved in the development of different tissues and organs, laying a foundation for future functional analysis of *JcRRs*.

The outgrowth of axillary buds was regulated by multiple interactions of hormones, nutritional conditions, and environmental factors. Type-B RRs, the critical component of CK signaling [68], played a vital role in the promotion of axillary bud outgrowth. Type-B *ARR1*, *ARR2*, *ARR10,* and *ARR12* synergistically affected shoot regeneration and axillary meristem development in *Arabidopsis* [69]. Their homologs were respectively identified as *JcRR1* and *JcRR12* in *J. curcas*, and we considered that these two genes may play critical roles in CK signaling. However, the regulatory mechanism of type-B RRs is complex, and a recent study found that *ARR1* has contrary function to *ARR10* in some cases [21]. We found that overexpressing *JcRR12* promoted decapitation-induced rosette branch outgrowth in Arabidopsis, which is consistent with the *arr1* mutant [22]. However, the regulatory network may be largely different between herbaceous and woody plants. Therefore, further studies of *JcRR12* in *J. curcas* are needed to elucidate the molecular mechanism underlying axillary bud outgrowth.

## 4. Materials and Methods

### 4.1. Plant Materials and Growth Conditions

Two-year-old *J. curcas* trees were planted in the field at Guangxi University, Nanning, Guangxi, China (22°82′ N, 108°32′ E; 100 m above sea level). Seedlings to be used for the 6-BA treatment were grown in the greenhouse (28 °C, 12 h light/12 h dark photoperiod, 70% humidity). Flowers, fruits, and seeds at different developmental stages were collected in May-June 2022 for qRT–PCR analysis. All other samples used for qRT–PCR experiments were collected at the same time in May 2022. Various plant tissue samples, including lateral roots 1–2 mm in diameter with fine roots and root tips, shoot apices 0.3 cm in length from the top of shoots, stems 1.5 cm in diameter, young leaf blades 2 cm in length, mature leaf blades 15 cm in length, the first axillary bud closest to the shoot apical meristem, recently opened female and male flowers, fruits at 15 days after pollination, and seeds 30 days after pollination, were harvested for qRT–PCR analysis. All the tissues were immediately frozen in liquid nitrogen and stored at −80 °C until needed.

Wild-type *Arabidopsis thaliana* ecotype Columbia (Col-0) and the transgenic lines were grown in an environmentally controlled room (22 °C, 16 h light/8 h dark photoperiod, 70% humidity).

### 4.2. Sequence Identification and Phylogenetic Tree Construction

The genome sequences and annotations were downloaded from the *J. curcas* database (JCDB) (http://jcdb.liu-lab.com/, accessed on 13 March 2022) [42], and we further downloaded the genome and annotations of the other three species for comparison in EnsemblPlants (http://EnsemblPlants.org/, accessed on 13 March 2022). The selection of RR members in *A. thaliana* [50], *P. trichocarpa* [16], and *O. sativa* [51] was obtained from published papers, and the sequences of RR members in each species are listed in Appendix A. Local BLASTP was used with the default parameters to search the genome sequence of *J. curcas*; the known Arabidopsis RR sequences were used as queries. The obtained candidate sequences were submitted to the Pfam database (http://pfam.xfam.org, accessed on 15 March 2022) for domain prediction. The sequences of *RR* genes in *J. curcas* containing the receiver domain (conserved domain of RR proteins) were ultimately retained. Phylogenetic trees of RR members in four species (Appendix A) were constructed by using MEGA 7.0 software (http://www.megasoftware.net (accessed on 15 March 2022)) with the NJ method and 1000 bootstrap replicates and the output was visualized using the online software tool iTOL (https://itol.embl.de/tree/, accessed on 4 June 2022).

### 4.3. Analysis of Protein Structure and Physicochemical Properties

Protein length, MW, theoretical pI, instability index, and GRAVY values of the JcRR proteins were calculated by using the ProtParam tool (https://web.exp-asy.org/protparam/, accessed on 13 June 2022) [70]. The prediction of subcellular localizations of JcRRs was analyzed by using Plant-mPLoc (http://www.csbio.sjtu.edu.cn/bioinf/plant-multi/, accessed on 13 June 2022) [71]. The results were later visualized by using TBtools (https://github.com/CJ-Chen/TBtools/releases (accessed on 13 March 2022)) [72].

### 4.4. Cis-Acting Elements in the Promoter Region and Gene Structure Analysis

The 1500 bp upstream coding sequence of *JcRRs* was obtained. The sequences were submitted to the PlantCARE website (http://bioinformat-ics.psb.ugent.be/web-t-ools/plantcare/html/, accessed on 12 June 2022) to analyze the *cis*-acting elements.

### 4.5. Conserved Motif and Domain Prediction

The conserved motifs of the JcRR proteins were analyzed using MEME (https://meme-suite.org/meme/tools/meme, accessed on 4 June 2022). The maximum number of motifs to be identified was set to 10. The conserved domains of JcRR family members were predicted by using the Batch Web CD-Search Tool (https://www.ncbi.nlm.nih.gov/Structure/bwrpsb/bwrpsb.cgi, accessed on 4 June 2022).

### 4.6. Chromosomal Distribution and Synteny Analysis

Chromosomal localization and synteny analysis between *J. curcas* and the other three species (Arabidopsis, Populus, and rice) were performed using TBtools (program “Advanced circos” and “One StepMC ScanX”). Then we used the “Simple Ka/Ks Calculator (NG)” program of TBtools to calculate the rates of synonymous substitutions (Ks), nonsynonymous substitutions (Ka), and evolutionary rates (Ka/Ks ratio) of the *J. curcas RR* gene family. Scatter plot for the Ka/Ks value of duplicated *JcRR* gene pairs was constructed using R (https://www.r-project.org/, accessed on 11 October 2020, version 4.0.3).

### 4.7. Hormone, Stress Response, and Spatiotemporal Expression Analysis of JcRRs According to Transcriptome Data

Transcriptome data of *J. curcas* were downloaded from the JCDB (accessed on 13 March 2022). The raw data were retrieved from the NCBI Sequence Read Archive (SRA) database (https://www.ncbi.nlm.nih.gov/ (accessed on 15 March 2022)), including those in response to hormones [39], from different tissues [43,44,45], from roots and leaves under drought stress [46], under salt stress [47] and from leaves after cold treatment [48] (Appendix A). The data were collated, and a matrix was constructed by Microsoft Office Excel (Appendix A). Heatmaps for the expression profiles of *JcRRs* were constructed using R. Data in heatmaps were normalized for each row. Hierarchical clustering was performed, and the default parameters were used.

### 4.8. Exogenous CK Treatment in J. curcas

Lovastatin (SL8280, Solarbio, Beijing, China), GA_3_ (G7645, Sigma, Saint Louis, USA), and 6-BA (B3408, Sigma) were first dissolved in 1 M NaOH, anhydrous ethanol, and dimethyl sulfoxide (DMSO) to form a stock solution (20 mM). The lovastatin stock solution was later diluted to 100 µM working solutions, while GA_3_ and 6-BA were diluted to 200 µM working solutions. For hormone treatments, 20 μL of the working solutions were directly applied to the leaf axil at node 1 of three-week-old *J. curcas* seedlings [39]. For GA_3_, lovastatin, and decapitation treatment, the control group was plants treated with the same concentration of ethanol or DMSO, or intact plants, respectively. The length of each axillary bud was measured with a vernier caliper at 14 d after treatment. For the 6-BA treatment, the seedlings were treated at 0 h, 3 h, 6 h, 12 h, 24 h, and 48 h before collection, and the control group was treated with the same concentration of NaOH at the same time. All the samples were collected at one time, and the stems of axillary buds (approximately 10–20 mg, 3–4 mm in length) were separately sliced and immediately frozen in liquid nitrogen for subsequent qRT–PCR analysis.

### 4.9. mRNA Extraction and qRT–PCR

Total RNA was extracted with a Plant RNA Kit (R6827, OMEGA, Norcross, USA), and cDNA was synthesized using HiScript III All-in-one RT SuperMix Perfect for qPCR (R333, Vazyme, Nanjing, China). Quantitative real-time PCR (qPCR) was performed using ChamQ Universal SYBR qPCR Master Mix (Q311, Vazyme) on a Roche Light Cycler 96 Real-Time PCR Detection System (Roche Diagnostics, Mannheim, Germany), and then the threshold cycle (Ct) values of each gene were collected. The combination of three marker genes *JcActin*, *JcGAPDH,* and *JcEF1ɑ* was used for the normalization of gene expression profiles for all developmental stages [73]. The primers used for qRT–PCR are listed in Appendix A. qRT–PCR was performed for three independent biological replicates (tissue samples were harvested from different plants) and three technical replicates per sample. The data were analyzed using the 2^−^^ΔΔCT^ method as described by Livak [74]. The calculation formula is shown below, where y is the expression level of *JcRRs*, and Ct_x1_ is the value of each *JcRR* used for the study. Ct_xA_, Ct_xG,_ and Ct_xE_ are the Ct values of *JcActin*, *JcGAPDH,* and *JcEF1ɑ*, respectively.
(1)$$y=2^{-\left[Ct_{x1}-\frac{1}{3}(Ct_{xA}+Ct_{xG}+Ct_{xE})\right]}$$

### 4.10. Construction of the JcRR12 Overexpression Vector and Arabidopsis Transformation

The CDS of *JcRR12* was amplified with the primers listed in Appendix A (*JcRR12-OE-F*/R). The product was cloned into the pEASY^®^-Blunt simple cloning vector (CB111, TransGen, Beijing, China). To generate the *35S:JcRR12* overexpression vector, the plant transformation vector pOCA30 [75] and the pEASY^®^-Blunt simple cloning vector containing the CDS of *JcRR12* were digested by *Xba*I/*Sal*I, and the resulting fragments were ligated by using T4 DNA ligase (EL0014, Thermo Scientific, Waltham, MA, USA). The generated *35S:JcRR12* plasmid was transformed into Arabidopsis plants using *Agrobacterium tumefaciens* EHA105. Transformation of Arabidopsis was performed using the floral dip method [76]. Transgenic plants were grown on 1/2 MS medium (PM1061, Coolabor, Beijing, China) containing 50 μg/mL kanamycin (CK6731, Coolabor). After identification, T3 generation homozygous transgenic plants were used for the statistical analysis of phenotypes.

### 4.11. Decapitation Assays and Branch Counts in Arabidopsis

For decapitation treatment, primary bolts were removed when the plants reached a height of 1 cm after the onset of flowering, and the control group was treated with intact plants. The number of rosette branches was investigated 10 days after treatment. For each group, at least 10 plants were investigated.

### 4.12. Statistical Analysis

The error bars in all figures indicate the standard deviation. SPSS 26.0 (SPSS, Inc., Chicago, IL, USA) was used to assess statistical significance by one-way ANOVA and Student’s *t*-test.

## 5. Conclusions

Exogenous treatment showed that CK was indispensable in GA-promoted axillary bud outgrowth, which indicates that GA may regulate axillary bud outgrowth by interacting with CK signaling. In addition, it also played a vital role in the release of apical dominance. To further elucidate the interaction mechanism, in this study, we identified 14 members of the RR gene family in *J. curcas*. The physicochemical characteristics, phylogenetic relationships, subcellular localization, and conserved structural domains of JcRR proteins and the chromosomal localization and gene structure of *JcRRs* were analyzed. Several vital *cis*-acting elements in the promoter region of the *JcRRs* were identified, and these elements are related to multiple hormones and stress responses.

The temporospatial expression patterns of *JcRRs* during the development of various tissues and their response to phytohormones and abiotic stress were further characterized via analysis of RNA-seq and qRT–PCR data. The results suggested that JcRRs are widely involved in phytohormone crosstalk and abiotic stress responses. *JcRRs* were mainly expressed in the roots, while they also showed differential expression patterns in other tissues, suggesting that *JcRRs* may also play vital roles in the development of other tissues in addition to roots. The preliminary functional analysis of *JcRR12* in Arabidopsis indicates its role in the decapitation-induced axillary bud outgrowth, which deserves further investigation in *J. curcas*. In conclusion, our study provides information for further functional studies of *JcRRs*, especially those that may be involved in axillary bud outgrowth, flower sex differentiation, and stress tolerance.

## Figures and Tables

**Figure 1 ijms-23-11388-f001:**
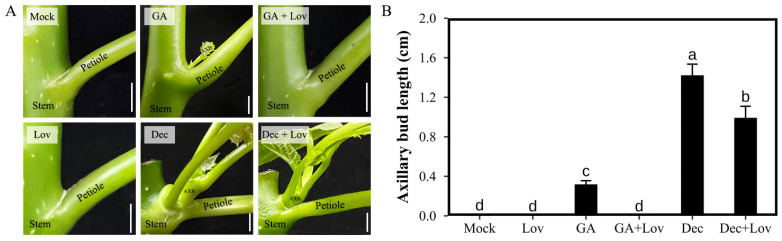
Lovastatin (Lov) inhibited the gibberellin A3 (GA_3_)- and decapitation-promoted outgrowth of axillary buds in 3-week-old *J. curcas* seedlings. (**A**) Treatment with 200 µM GA_3_ and decapitation promoted the outgrowth of axillary buds, while 100 µM Lov inhibited these processes. (**B**) The elongation of axillary buds under different treatments. AXB, axillary bud. Scale bars = 1 cm. All data and photos were taken at 2 weeks after treatment. Different letters indicate significant differences (*p* < 0.01, *n* = 10, one-way ANOVA, Tukey’s test).

**Figure 2 ijms-23-11388-f002:**
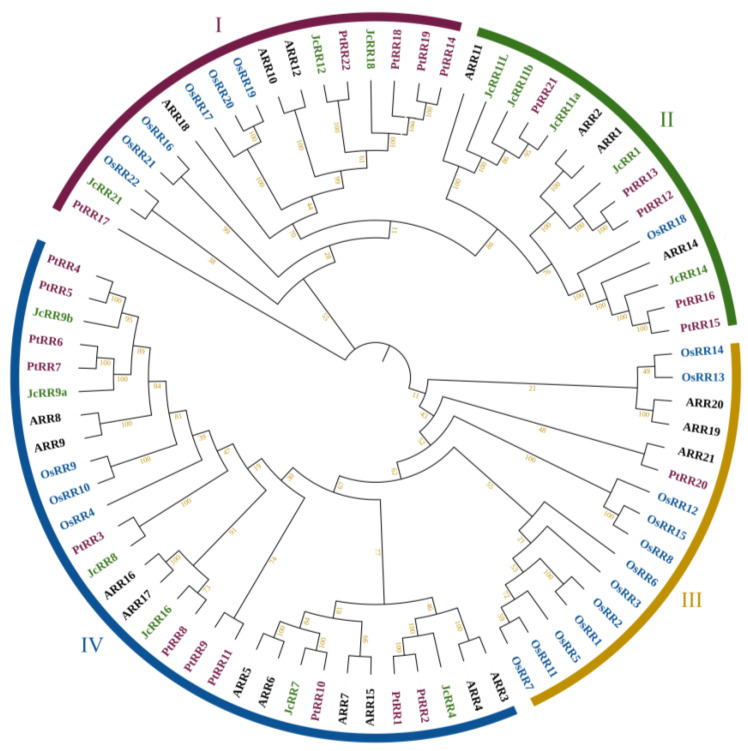
Phylogenetic tree of RR proteins among *J. curcas* (*Jc*), Arabidopsis, Populus, and rice. The unrooted phylogenetic tree was constructed via MEGA 7.0 by the neighbor-joining method with 1000 bootstrap replicates. The RRs were divided into four major subfamilies. The different subfamilies are indicated with different colors. The green, black, purple and blue words represent *J. curcas*, Arabidopsis, Populus and rice, respectively.

**Figure 3 ijms-23-11388-f003:**
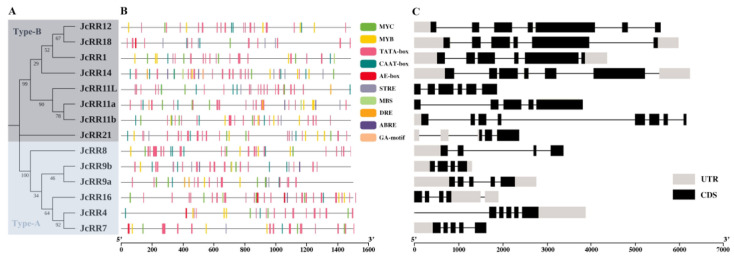
Evolutionary analysis and genomic organization of *JcRRs*. (**A**) Phylogenetic tree of type-A and type-B *JcRRs*. (**B**) Analysis of *cis*-acting elements in the promoter region (1500 base pairs (bp) upstream of the start codon); each motif is represented by a specific color. (**C**) Genomic organization of *JcRRs*. Untranslated regions (UTRs) and coding DNA sequence (CDS) regions are represented by solid gray and black boxes, respectively.

**Figure 4 ijms-23-11388-f004:**
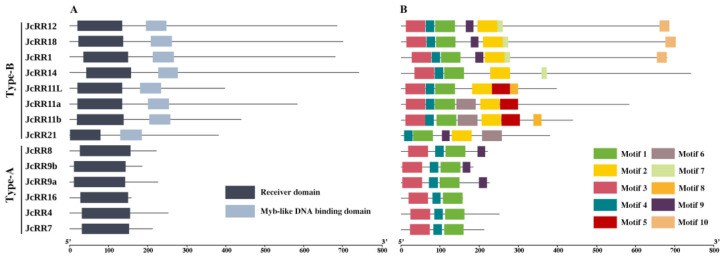
Analysis of the conserved domain and motif elements of the *RR* gene family members in *J. curcas*. (**A**) Domain structure of RRs; (**B**) Motif composition of JcRRs. Each motif is represented by a specific color.

**Figure 5 ijms-23-11388-f005:**
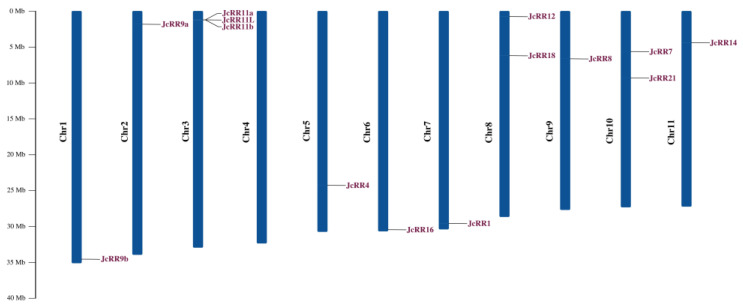
Locations and duplications of putative RRs on *J. curcas* chromosomes. Shown is the distribution of *RR* genes across the chromosomes of *J. curcas* plants. The genetic distance of seven chromosomes is represented by the scale in megabases (Mb) on the left. The *JcRRs* are displayed on the basis of the nomenclature used for the high-quality reference genome of *J. curcas*. The black lines represent the location of the genes on each chromosome.

**Figure 6 ijms-23-11388-f006:**
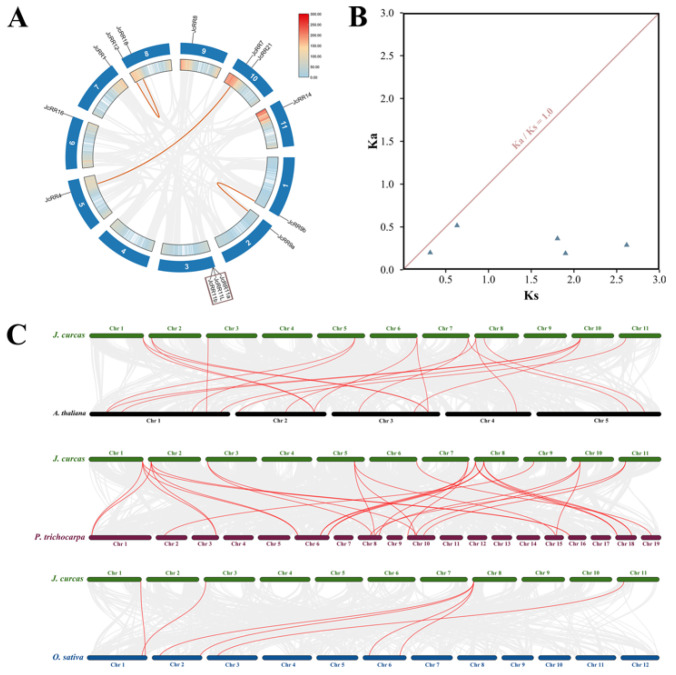
Evolutionary analysis of RRs. (**A**) Gene density and duplications of putative RRs on *J. curcas* chromosomes. The heatmap represents the density of genes at corresponding positions on the chromosome. Collinear blocks in the whole *J. curcas*, and red lines indicate duplicated *JcRR* gene pairs. The dark red rectangle represents tandem duplications. (**B**) Ka/Ks values of duplicated *JcRR* gene pairs. Five pairs of duplicated *JcRR* genes with Ka/Ks less than one. The detailed Ka/Ks information of duplication pairs is described in Appendix A. (**C**) Collinearity analysis of *J. curcas* with three other species (*A. thaliana*, *P. trichocarpa* and *O. sativa*). Chr, the serial number of each chromosome. The gray lines in the background indicate the collinear blocks between *J. curcas* and other plant genomes, while the red lines highlight the syntenic *RR* gene pairs.

**Figure 7 ijms-23-11388-f007:**
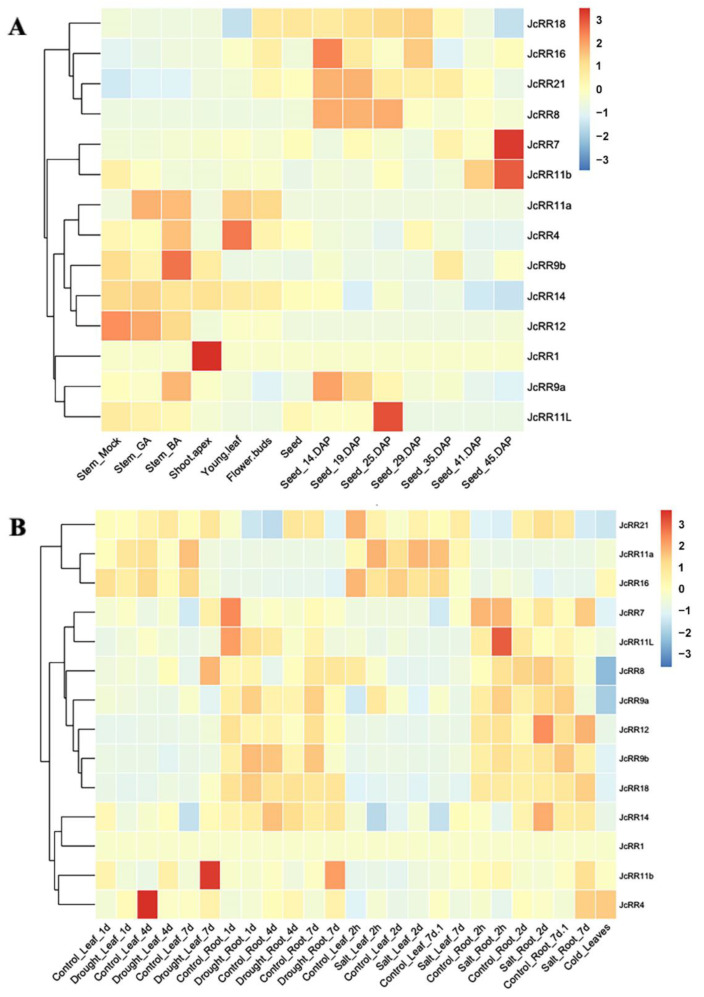
Expression patterns of *JcRRs*, as revealed by RNA sequencing analysis. (**A**) Expression in response to 6-benzylaminopurine (6-BA) or GA_3_ treatment, in different tissues, and in seeds at different developmental stages after pollination. (**B**) Leaves and roots under different stresses. The genome-wide expression profiles of *JcRRs* are shown on a heatmap using the log_2_ (reads per kilobase per million mapped reads (RPKM)) value, and −3.00 to 3.00 were artificially set with the color scale limits according to the normalized value. The color scale shows increasing expression levels from blue to red.

**Figure 8 ijms-23-11388-f008:**
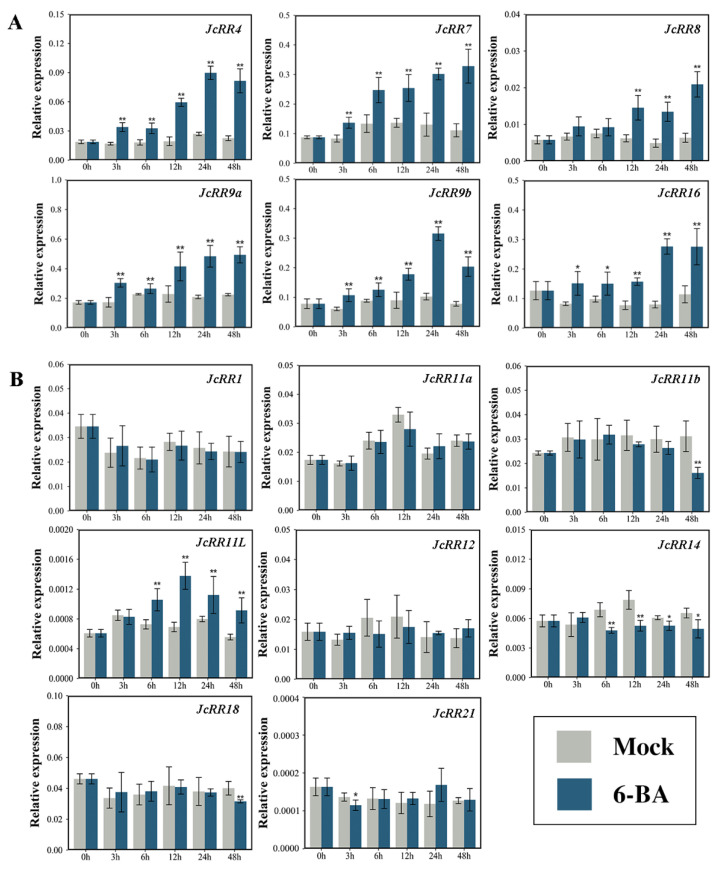
Expression profiles of *JcRRs* in response to exogenous 6-BA treatment, as revealed by qRT–PCR analysis. (**A**) Expression profiles of 6 type-A *JcRRs*. (**B**) Expression profiles of 8 type-B *JcRRs*. Samples were collected at 0 h, 3 h, 6 h, 12 h, 24 h, and 48 h after 6-BA (200 µM) treatment. The mock group served as a control. Student’s *t*-test was used to determine significant differences between the experimental group and control group. Significance levels: *, *p* < 0.05; **, *p* < 0.01.

**Figure 9 ijms-23-11388-f009:**
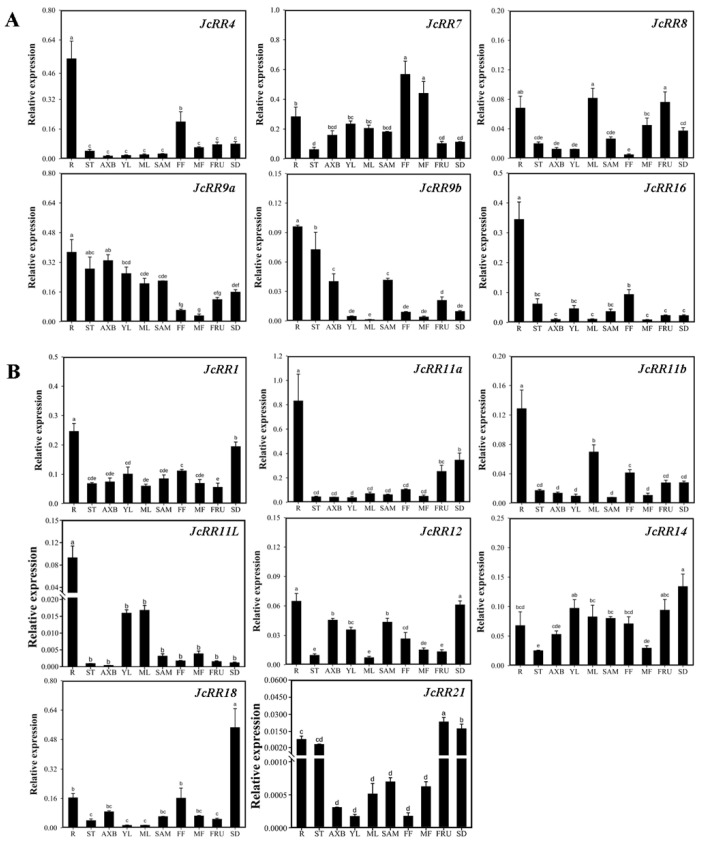
Expression profiles of *JcRRs* in various tissues, as revealed by qRT–PCR analysis. (**A**) Expression profiles of 6 type-A *JcRRs*. (**B**) Expression profiles of 8 type-B *JcRRs*. qRT–PCR analysis was performed on ten tissues: R, roots; ST, stems; AXB, axillary buds; YL, young leaves; ML, mature leaves; SAM, shoot apical meristems; FF, female flowers; MF, male flowers; FRU, fruits; SD, seeds. The qRT–PCR results were obtained from three independent biological replicates and three technical replicates per sample. The error bars represent the standard deviations (SDs). Different letters indicate significant differences (*p* < 0.05, *n* = 3, one-way ANOVA, Tukey’s test).

**Figure 10 ijms-23-11388-f010:**
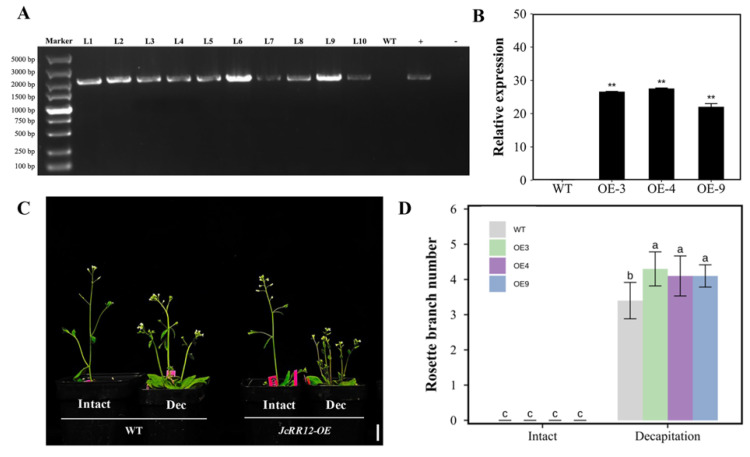
Phenotypic changes in *35S:JcRR12* transgenic *Arabidopsis*. (**A**) PCR identification of *35S:JcRR12* transgenic *Arabidopsis*. WT, wild type negative control; +, plasmid positive control; −, ddH_2_O negative control. (**B**) qRT-PCR analysis of *JcRR12* expression in WT and transgenic *Arabidopsis* (OE-3, OE-4 and OE-9). The qRT-PCR results were obtained from three biological replicates and three technical replicates. The values were normalized to the expression of *AtActin2*. The error bars represent the standard deviations (SDs). Student’s *t*-test was used to determine significant differences between transgenic and WT Arabidopsis. Significance levels: **, *p* < 0.01. (**C**) Phenotypes of WT and transgenic *Arabidopsis* after decapitation (Dec) treatment. Scale bar = 1 cm. (**D**) Number of rosette branches of WT and transgenic *Arabidopsis* after mock, 6-BA and decapitation treatments. The values are presented as the means ± standard deviations. Different letters indicate significant differences (*p* < 0.05, *n* = 10, one-way ANOVA, Tukey’s test).

**Table 1 ijms-23-11388-t001:** Features of cytokinin response regulators in *J. curcas*. GRAVY, grand average of hydropathicity. kDa, kilodaltons.

Name	Gene ID	Type	Protein Length (aa)	MW (kDa)	pI	Subcellular Localization	Instability Index	GRAVY
JcRR4	105649825	A	252	27.79	8.18	Nucleus	87.84	−0.392
JcRR7	105640448	A	212	23.60	8.28	Nucleus	66.37	−0.248
JcRR8	105638952	A	222	25.06	4.86	Nucleus	69.70	−0.817
JcRR9a	105634414	A	226	25.52	5.80	Nucleus	74.07	−0.733
JcRR9b	110009224	A	185	20.90	5.19	Nucleus	53.56	−0.523
JcRR16	105649171	A	158	17.77	7.58	Nucleus	31.31	−0.313
JcRR1	105643496	B	680	74.15	5.64	Nucleus	44.46	−0.459
JcRR11a	105644735	B	583	65.66	5.65	Nucleus	45.23	−0.618
JcRR11b	105644733	B	439	49.37	5.76	Nucleus	48.08	−0.608
JcRR11L	105644734	B	398	44.86	6.52	Nucleus	36.01	−0.387
JcRR12	105637574	B	685	75.29	5.79	Nucleus	32.63	−0.488
JcRR14	105634128	B	741	80.28	6.27	Nucleus	44.31	−0.272
JcRR18	105633481	B	701	76.85	5.47	Nucleus	43.80	−0.566
JcRR21	110009834	B	381	43.11	5.55	Nucleus	45.77	−0.671

## Data Availability

Not applicable.

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
