# Peer review of "Genome-Wide Identification and Expression Analysis of Cytokinin Response Regulator (RR) Genes in the Woody Plant Jatropha curcas and Functional Analysis of JcRR12 in Arabidopsis"

_ijms, 2022, doi:10.3390/ijms231911388_

Round 1
Reviewer 1 Report
The manuscript entitled “Genome-Wide Identification and Expression Analysis of Cyto-
kinin Response Regulator Genes in the Woody Plant Jatropha curcas” by Geng and collaborators presents the identification of genes related to the response to cytokinin (CK), in particular the transducers of the ARR family, as well as the expression characterization of these genes in seed of "curcas" in response to different stresses. The identification of different genes related to the phytohormones responses is a interesting point in plant genetics and physology. However, the current manuscript presents different flaws in the methodology, in addition to not presenting anything innovative regarding this group of genes. In the opinion of this reviewer, the study should be more ambitious and not only characterize the CK response genes, but also all known genes related to CK biosynthesis, signal translation and degradation. In this way, it is possible to have a broader vision of the metabolism of this important hormone in this plant of biotechnological interest. In addition, the authors must improve the following points for future publication.
From the point of view of this reviewer, the following points have to be taken into account, by the authors,
Line 96-99. The correct identification of genes cannot be done by blastp, being such distant species. For the correct identification of the genes, it is necessary to carry out an orthology study, using the complete proteomes of A. thaliana, P. trichocarpa and O. sativa, together with specific software such as Proteinortho (https://www.bioinf.uni-leipzig.de/Software/proteinortho/), or any methodology that allows identifying orthologs and paralogs. In all cases that use Blastp, they must indicate for example the cutoff of e-value, the number of hits, the database used, etc...
Line 372-373. It is necessary to include a supplementary table with the name and accession number of the A. thaliana, P. trichocarpa and O. sativa genes used. It is very necessary to know which genes have been used and which is their reference in a database accessible to the scientific community
Line 372 and 401. The from the JCDB database cannot be accessed using a normal browser, it is necessary to install the frp software, it is necessary that the data used is available from any browser. Therefore, it is difficult to verify the data indicated by the authors.
Line 378-381. Why do they use NJ for the phylogenetic model? What model do the authors use for reconstruction? They have carried out some test to verify that this is the best phylogenetic model for that group of genes. It is again necessary to indicate the accession number of a reference base of the genes used. In addition, they have to explain what they use for phylogenetic reconstruction, whether the transcripts or their translation into proteins. If so, they should include the sequences as supplementary material.
Line 401-408. The authors must indicate how they have processed the RNA-Seq data, since it is key to interpreting the results shown. In addition, the authors do not indicate how they perform the statistical analyses. They should be clearer on this part. Again, it is not possible to access and verify the data deposited in JCDB database
Figure 7. They have not carried out any statistical test, therefore, everything derived from this analysis cannot be confirmed as true. They must also indicate how they have carried out the statistical analysis in materials and methods.
From the point of view of this reviewer, the presented MS needs to correct and clarify all the indicated methodological points. In addition, it should include more genes in the study. For these reasons, in the opinion of this reviewer, the MS does not present sufficient quality for publication in IJMS.
Author Response
Response to Reviewer 1 Comments
We are grateful for your review and appreciate your suggestions on our manuscript. According to your suggestions, we have made modifications, and the following are the corresponding answers to each of your questions.
Point 1: In the opinion of this reviewer, the study should be more ambitious and not only characterize the CK response genes, but also all known genes related to CK biosynthesis, signal translation and degradation. In this way, it is possible to have a broader vision of the metabolism of this important hormone in this plant of biotechnological interest.
Response 1: Thank you for your suggestions. Cytokinin (CK) biosynthesis (JcIPTs and JcCYP735A) and degradation (JcCKXs) genes in Jatropha curcas have been characterized in our previous study (Cai et al., 2018). In this study, we focus on the CK response regulator genes. As we mentioned in the introduction of the manuscript (Lines 81-93 on page 2), CK response regulator genes play vital roles in plant development and responses to diverse stresses. The crosstalk between CK and other factors (other phytohormones, stress, etc.) are well known, but most of the mechanisms are still unclear. We found that gibberellins (GA) have a synergistic effect with CK in regulating axillary bud growth, so we investigated the signal transduction pathway of cytokinin to elucidate the mechanism.
Cai, L.; Zhang, L.; Fu, Q.; Xu, Z.F. (2018) Identification and expression analysis of cytokinin metabolic genes IPTs, CYP735A and CKXs in the biofuel plant Jatropha curcas. PeerJ, 6, e4812.
Point 2: Line 96-99. The correct identification of genes cannot be done by blastp, being such distant species. For the correct identification of the genes, it is necessary to carry out an orthology study using the complete proteomes of A. thaliana, P. trichocarpa and O. sativa, together with specific software such as Proteinortho (https://www.bioinf.uni-leipzig.de/Software/proteinortho/) or any methodology that allows identifying orthologs and paralogs. In all cases that use Blastp, they must indicate for example the cut off of e-value, the number of hits, the database used, etc...
Response 2: Thank you for your suggestions. The approach you suggest is very professional, and we are sorry that we do not offer such an analysis. Because response regulator (RR) members of three species (Arabidopsis, Populus and rice) had been identified and published, we downloaded corresponding genomic sequences from EnsemblPlants (http://EnsemblPlants.org/) and selected the corresponding RR members in Text S1-S4. Arabidopsis ARR sequences were used to query the J. curcas proteomes. In the first round of blastP, we set the expected threshold as 10, and we obtained 40 candidates. Then, we performed structural domain screening based on the Pfam database. In addition, we also analyzed the conserved D-D-K receiver domain and Myb-like DNA domain and motifs (Figure 4A) to ensure correct identification of JcRR genes from J. curcas. Then, we generated a multiple sequence alignment to check the matching degree among ARRs and JcRRs again. Finally, we identified 14 JcRR members according to the conserved D-D-K receiver domain of RR proteins. Moreover, we performed a reverse blastP to query Arabidopsis proteomes using 14 JcRR protein sequences we obtained. We also carried out a collinearity study to show the orthologous relationship between JcRRs and RRs from other species. Thus, we think the identification of JcRRs by this combination analysis is also convincing. More detailed information has been added to the ‘Materials and Methods’ and supplementary Tables in the revised manuscript. Hopefully, this may address your concerns.
Point 3: Line 372-373. It is necessary to include a supplementary table with the name and accession number of the A. thaliana, P. trichocarpa and O. sativa genes used. It is very necessary to know which genes have been used and which is their reference in a database accessible to the scientific community.
Response 3: Thank you for your suggestions. We added supplementary Table S1-S4 to the revised manuscript, which includes the name, type, accession number and sequences of all RR members of A. thaliana, P. trichocarpa, and O. sativa used in this study, respectively. For convenience, we use the Gene ID in the National Center for Biotechnology Information (NC-BI) (https://www.ncbi.nlm.nih.gov/) as the accession number of each sequence.
Point 4: Line 372 and 401. The JCDB database cannot be accessed using a normal browser, it is necessary to install the frp software, it is necessary that the data used is available from any browser. Therefore, it is difficult to verify the data indicated by the authors.
Response 4: Thank you for your suggestion. We have resolved issues with database access, and the website is now accessible. In addition, we also attached the data source information (Table S4) and normalized RNA-seq data used for heatmaps (Table S5) in the revised manuscript.
Point 5: Line 378-381. Why do they use NJ for the phylogenetic model? What model do the authors use for reconstruction? They have carried out some test to verify that this is the best phylogenetic model for that group of genes. It is again necessary to indicate the accession number of a reference base of the genes used. In addition, they have to explain what they use for phylogenetic reconstruction, whether the transcripts or their translation into proteins. If so, they should include the sequences as supplementary materials.
Response 5: Thank you for your questions. We used five methods for phylogenetic construction (UPGMA, ME, NJ, MP and ML), and we obtained the same results. Since the NJ method is more commonly used, we used this method for the phylogenetic model. We have added the accession number and corresponding protein sequences for each gene used in the construction of the phylogenetic tree in Text S1-S4 in the revised manuscript.
Point 6: Line 401-408. The authors must indicate how they have processed the RNA-Seq data, since it is key to interpreting the results shown. In addition, the authors do not indicate how they perform the statistical analyses. They should be clearer on this part. Again, it is not possible to access and verify the data deposited in JCDB database
Response 6: Thank you for your suggestions. Our previous study (Zhang et al., 2019) processed the RNA-seq data into a matrix table with FPKM values of different genes in Jatropha curcas, which were uploaded to the JCDB website. We filtered and organized the data for JcRRs (Table S4-S5). Then, we generated heatmaps for the visualization of the relative expression levels of JcRRs under different conditions. There were multiple variable groups, so it was difficult to mark significance in the heatmaps. We further performed qRT-PCR analysis to verify RNA-Seq data, and added statistical analysis in Figure 8 and Figure 9. We are very sorry that the website was not accessible when you visited, but now issues with JCDB access has been resolved.
Zhang, X.; Pan, B.-Z.; Chen, M.; Chen, W.; Li, J.; Xu, Z.-F.; Liu, C. (2019). JCDB: a comprehensive knowledge base for Jatropha curcas, an emerging model for woody energy plants, BMC Genomics, 20, 958.
Point 7: They have not carried out any statistical test, therefore, everything derived from this analysis cannot be confirmed as true. They must also indicate how they have carried out the statistical analysis in materials and methods.
Response 7: Thank you for your suggestions. The missing statistical analysis information has been added to Figure 9 (corresponding to the old Figure 7) and in ‘Materials and Methods’ section in the revised manuscript (lines 726-729 in page 17).

Reviewer 2 Report
In the manuscript entitled “Genome-Wide Identification and Expression Analysis of Cytokinin Response Regulator Genes in the Woody Plant Jatropha curcas” by Geng, et al., authors have conducted a genome-wide identification and expression analysis of cytokinin (CK) response regulator (RR) gene family in Jatropha curcas. The major drawback of the study is lack of creative work, also there are several flaws in the present format of manuscript.
- In figure 6, the results presented in the text is contradictory to the data presented in the graphs.
- Statistical analysis is not there in figure 7.
- Authors have used Actin as an endogenous control for normalization of qRT-PCR data, I was wondering if the authors have performed any analysis to check the stable expression of act during stress treatment.
- Additionally, authors should use at least two endogenous control for the calibration of qRT-PCR data to obtain a firm and authenticate results. As, it is generally not acceptable to only use one marker gene to normalize gene expression (Guenin et al., 2009 J. Exp. Bot. 60: 487–493, Dekkers et al., 2012 Plant Cell Physiol. 53(1):28–37; Salvi et al., 2017 Plant Cell Physiol). Besides, there is a previous report which document the stability of reference genes in Jatropha under different physiological conditions (Zhang et al., 2013; Karuppaiyaet al., 2017)
Zhang, L., He, L. L., Fu, Q. T., & Xu, Z. F. (2013). Selection of reliable reference genes for gene expression studies in the biofuel plant Jatropha curcas using real-time quantitative PCR. International journal of molecular sciences, 14(12), 24338-24354.
Karuppaiya, P., Yan, X. X., Liao, W., Wu, J., Chen, F., & Tang, L. (2017). Identification and validation of superior reference gene for gene expression normalization via RT-qPCR in staminate and pistillate flowers of Jatropha curcas–A biodiesel plant. PloS one, 12(2), e0172460.
- Authors have mentioned that they have used △△CT method for qrt analysis, however they not clearly explained the methodology how they have analysed the qRT data. Authors should have elaborated how they have done the relative expression analysis? how they have done the calibration? I would request authors to reanalyse the data presented.
- The evolutionary assessment is also missing.
- Overall, it is a genome wide identification and expression profiling, which is not a good fit for this journal.
Author Response
Response to Reviewer 2 Comments
We are grateful for your review and appreciate your suggestions on our manuscript. According to your suggestions, we have made modifications, and the following are the corresponding answers to each of your questions.
Point 1: In figure 6, the results presented in the text is contradictory to the data presented in the graphs.
Response 1: Thank you for your suggestions. The error in the legend of old Figure 6 (now named Figure 8) has been corrected in the revised manuscript.
Point 2: Statistical analysis is not there in figure 7.
Response 2: Thank you for your suggestions. The missing significance analysis has been added to old Figure 7 (now named Figure 9). The methods of statistical analysis are added in the figure legend and in the ‘Materials and Methods’ section (lines 415-417 in page 11, and lines 726-729 in page 17).
Point 3: Authors have used Actin as an endogenous control for normalization of qRT-PCR data, I was wondering if the authors have performed any analysis to check the stable expression of act during stress treatment.
Response 3: Thank you for your question. The qRT-PCR analysis in this study did not involve the data under stress treatment. The transcriptome data during stress treatment were normalized in a previous study (Zhang et al., 2019).
Zhang, X.; Pan, B.-Z.; Chen, M.; Chen, W.; Li, J.; Xu, Z.-F.; Liu, C. (2019). JCDB: a comprehensive knowledge base for Jatropha curcas, an emerging model for woody energy plants, BMC Genomics, 20, 958.
Point 4: Additionally, authors should use at least two endogenous controls for the calibration of qRT-PCR data to obtain a firm and authenticate results. As, it is generally not acceptable to only use one marker gene to normalize gene expression.
Response 4: Thank you for your suggestions. According to a previous study (Zhang et al., 2013), which ranked the stability of different marker genes in Jatropha curcas, we re-performed the qRT-PCR analysis and normalized the expression levels of JcRRs by using the combination of three marker genes JcActin, JcGAPDH, and JcEF1α. The corresponding information has been revised in the revised manuscript (lines 673-682 in page 16).
Zhang, L.; He, L.-L.; Fu, Q.-T.; Xu, Z.-F., Selection of reliable reference genes for gene expression studies in the biofuel plant Jatropha curcas using real-time quantitative PCR. Int. J. Mol. Sci. 2013, 14, 24338-24354.
Point 5: Authors have mentioned that they have used △△CT method for qrt analysis, however they not clearly explained the methodology how they have analysed the qRT data. Authors should have elaborated how they have done the relative expression analysis? how they have done the calibration? I would request authors to reanalyze the data presented.
Response 5: Thank you for your suggestions. We have added more detailed information in ‘Materials and Methods’ (lines 670-679 on page 16) about the calculation method of qRT-PCR data in the revised manuscript.
Point 6: The evolutionary assessment is also missing.
Response 6: Thank you for your suggestions. We have added this missing assessment in ‘2.5. Chromosomal Distribution and Synteny Analysis’ (lines 251–257 on page 6, and lines 286-301 on page 7) as well as new Figure 6 in the revised manuscript.
Point 7: The major drawback of the study is lack of creative work
Response 7: Thank you for your comment. We believe that the identification and expression analysis of JcRRs will help us to carry out further functional studies of JcRRs. To improve the quality of our study, we added additional results to the revised manuscript, including the interaction between CK and other factors in regulation of axillary bud outgrowth (lines 126-145 on page 3) and a preliminary functional study of the JcRR12 gene in Arabidopsis (lines 415-423 on page 11, and lines 424-440 on page 12). These results will help us to further explore the mechanism underlying the synergistic regulation of axillary bud growth by CK and GA in J. curcas.

Reviewer 3 Report
In this work, the authors performed bioinformatic and expression analyses to characterize the CK RR gene (JcRR) family in woody plant J. curcas, and provided a detailed knowledge of JcRRs for further analysis of CK signaling and JcRR functions in J. curcas. The work is of great significance as an example in woody plants, and I would like recommend acceptance after a minor revision.
Abstract:
Q1,lines 20-21:Could the sentence be: “The Myb-like DNA-binding domain divides the 14 members of the JcRR family into type A and type B proteins.” ?
Q2, lines 23:Not only plant hormones, light and abiotic stress factors in plant development, there should be other factors.This conclusion is too absolute.
Q3, lines 25:“we characterized the temporospatial expression…” should be “we characterized that…”.
Q4, lines 28-29:“The expression levels of all 6 type-A and 1 type-B JcRRs increased in response to 6-BA…”.6-BA is the first appearance, should write the full name.
Keywords:
Q5:Please, put the keywords in an alphabetical order.
Introduction:
Q6: lines 42-43:Could the author explain the TCS system in detail?
Q7, lines 90: “we determined…” should be “we determined that…”.
Results:
Q8, lines 96-97:Could the sentence be: “Because all the RR members in Arabidopsis, Populus and rice have been identified, and their functions have been characterized, sequences of them were selected to query the J. curcasgenome.”?
Q9, lines 101-102:“There were significantly fewer JcRRs than Arabidopsis RR homologues, so we named these JcRRs according to their homologs in Arabidopsis”.Could the author explain this phenomenon in detail?
Q10, Table 1:Could the author explain in detail what JcRR11a, JcRR11b, JcRR11L stand for ? What are their differences ?
Q11, lines 124-128:The plant names in sentences should be unified.
Q12, lines 142-143:“The promoter region of the JcRRs contained multiple cis-acting elements, including phytohormone-responsive, light-responsive, plant development-related, and stress-responsive elements.”The author should list the corresponding components.
Q13, lines 146-147:“The structures of the JcRRs were relatively different, which included different numbers and positions of exons and introns”. Could the author explain this phenomenon in detail ?
Q14, lines 183-190: The content described here is too lengthy.
Others to check
Author Response
Response to Reviewer 3 Comments
We are grateful for your approval and appreciate your suggestions on our manuscript. According to your suggestions, we have made serious modifications, and the following are the corresponding answers to each of your questions.
Point 1: Lines 20-21: Could the sentence be: “The Myb-like DNA-binding domain divides the 14 members of the JcRR family into type A and type B proteins.”?
Response 1: Thank you for your suggestions. This sentence has been edited in the revised manuscript (lines 24-25 on page 1).
Point 2: Lines 23: Not only plant hormones, light and abiotic stress factors in plant development, there should be other factors. This conclusion is too absolute.
Response 2: Thank you for your suggestions. This conclusion has been revised (line 27 on page 1).
Point 3: Lines 25: “we characterized the temporospatial expression…” should be “we characterized that…”.
Response 3: Thank you for your suggestions. The sentence has been edited in the revised manuscript (line 31 on page 1).
Point 4: Lines 28-29: “The expression levels of all 6 type-A and 1 type-B JcRRs increased in response to 6-BA…”.6-BA is the first appearance, should write the full name.
Response 4: Thank you for your suggestions. This missing information has been added to the revised manuscript (line 35 on page 1).
Point 5: Please, put the keywords in an alphabetical order.
Response 5: Thank you for your suggestions. The keywords have been listed alphabetically in the revised manuscript (line 55 on page 2).
Point 6: Lines 42-43: Could the author explain the TCS system in detail?
Response 6: Thank you for your suggestions. We have added a detailed explanation to the revised manuscript (lines 62-67 on page 2).
Point 7: Lines 90: “we determined…” should be “we determined that…”.
Response 7: Thank you for your suggestions. We have modified this sentence in the revised manuscript (line 121 on page 3).
Point 8: Lines 96-97: Could the sentence be: “Because all the RR members in Arabidopsis, Populus and rice have been identified and their functions have been characterized, sequences of them were selected to query the J. curcas genome.”?
Response 8: Thank you for your suggestions. We have modified this sentence in the revised manuscript (lines 147-149 on page 3).
Point 9: Lines 101-102: “There were significantly fewer JcRRs than Arabidopsis RR homologues, so we named these JcRRs according to their homologs in Arabidopsis”. Could the author explain this phenomenon in detail?
Response 9: Thank you for your suggestions. We have added a more detailed explanation of this phenomenon in the revised manuscript (lines 173-176 on page 4).
Point 10: Table 1: Could the author explain in detail what JcRR11a, JcRR11b, JcRR11L stand for? What are their differences?
Response 10: Thank you for your questions. All three of them are closely related to Arabidopsis ARR11 (Figure 1), but JcRR11a and JcRR11b are clustered under the same branch, while JcRR11L is parallel to this branch (Figure 3A). The three genes mainly differ in genome structures, such as the number of introns and the length of introns and exons. In addition, the whole genomic DNA length of JcRR11L is significantly shorter than those of JcRR11a and JcRR11b (Figure 3C). These three genes had undergone two tandem replications (JcRR11a versus JcRR11L, and JcRR11L versus JcRR11b), and they have a relatively close distribution on chromosome 3 (Figure 5, Figure 6A, and Table S3).
Point 11: Lines 124-128: The plant names in sentences should be unified.
Response 11: Thank you for your suggestions. We have unified the plant names in the revised manuscript (line 202 on page 5).
Point 12: Lines 142-143: “The promoter region of the JcRRs contained multiple cis-acting elements, including phytohormone-responsive, light-responsive, plant development-related, and stress-responsive elements.” The author should list the corresponding components.
Response 12: Thank you for your suggestions. This missing element information has been added to the revised manuscript (lines 226-228 on page 6).
Point 13: Lines 146-147: “The structures of the JcRRs were relatively different, which included different numbers and positions of exons and introns”. Could the author explain this phenomenon in detail?
Response 13: Thank you for your questions. This missing information has been added to the revised manuscript (lines 231-234 on page 6).
Point 14: Lines 183-190: The content described here is too lengthy.
Response 14: Thank you for your suggestions. We have edited the content of the description in a concise way (lines 319-320 on page 8) in the revised manuscript.

Round 2
Reviewer 1 Report
The new version of MS “Genome-Wide Identification and Expression Analysis of Cyto-kinin Response Regulator (RR) Genes in the Woody Plant Jatropha curcas and Functional Analysis of JcRR12 in Arabidopsis” presents a series of changes suggested by the different reviewers, which allow compliance with the standards of the IJMS journal. From the point of view of this reviewer, the article can be published in its current state.
Reviewer 2 Report
The authors have addressed the concern raised in previous submission. The present format of manuscript can be accepted for publication.